# Exploring Mannosylpurines as Copper Chelators and Cholinesterase Inhibitors with Potential for Alzheimer’s Disease

**DOI:** 10.3390/ph16010054

**Published:** 2022-12-30

**Authors:** Ignazio Schino, Mariangela Cantore, Modesto de Candia, Cosimo D. Altomare, Catarina Maria, João Barros, Vasco Cachatra, Patrícia Calado, Karina Shimizu, Adilson A. Freitas, Maria C. Oliveira, Maria J. Ferreira, José N. C. Lopes, Nicola A. Colabufo, Amélia P. Rauter

**Affiliations:** 1Centro de Química Estrutural, Institute of Molecular Sciences, Faculdade de Ciências, Universidade de Lisboa, Ed C8, Piso 5, Campo Grande, 1749-016 Lisboa, Portugal; 2Biofordrug Srl, Via Dante 95, 70019 Triggiano, Italy; 3Department of Pharmacy-Pharmaceutical Sciences, University of Bari Aldo Moro, Via E. Orabona 4, 70125 Bari, Italy; 4Departamento de Química, Faculdade de Ciências e Tecnologia, Universidade Nova de Lisboa, Largo da Torre, 2825-149 Caparica, Portugal; 5Centro de Química Estrutural, Institute of Molecular Sciences Instituto Superior Técnico, Universidade de Lisboa, Av. Rovisco Pais, 1049-001 Lisboa, Portugal

**Keywords:** cholinesterase inhibitors, copper chelators, purine nucleosides, Alzheimer’s disease

## Abstract

Alzheimer’s Disease (AD) is characterized by a progressive cholinergic neurotransmission imbalance, with a decrease of acetylcholinesterase (AChE) activity followed by a significant increase of butyrylcholinesterase (BChE) in the later AD stages. BChE activity is also crucial for the development of Aβ plaques, the main hallmarks of this pathology. Moreover, systemic copper dyshomeostasis alters neurotransmission leading to AD. In the search for structures targeting both events, a set of novel 6-benzamide purine nucleosides was synthesized, differing in glycone configuration and N^7^/N^9^ linkage to the purine. Their AChE/BChE inhibitory activity and metal ion chelating properties were evaluated. Selectivity for human BChE inhibition required N^9^-linked 6-deoxy-α-d-mannosylpurine structure, while all three tested β-d-derivatives appeared as non-selective inhibitors. The N^9^-linked l-nucleosides were cholinesterase inhibitors except the one embodying either the acetylated sugar or the *N*-benzyl-protected nucleobase. These findings highlight that sugar-enriched molecular entities can tune bioactivity and selectivity against cholinesterases. In addition, selective copper chelating properties over zinc, aluminum, and iron were found for the benzyl and acetyl-protected 6-deoxy-α-l-mannosyl N^9-^linked purine nucleosides. Computational studies highlight molecular conformations and the chelating molecular site. The first dual target compounds were disclosed with the perspective of generating drug candidates by improving water solubility.

## 1. Introduction

Alzheimer’s Disease (AD) is a devastating disease with a multifactorial pathogenesis. The main histopathological hallmarks of AD are the generation and deposition of senile plaques containing extracellular aggregates of neurotoxic amyloid-β (Aβ) peptide, along with intracellular neurofibrillary tangles of hyperphosphorylated τ-protein and the impairment of the cholinergic neurotransmission, due to a decrease of the level of the neurotransmitter acetylcholine (ACh), leading to a degeneration of cholinergic neurons and neuronal loss [1,2]. In physiological conditions, ACh is predominantly hydrolyzed by acetylcholinesterase (AChE), while butyrylcholinesterase (BChE) has a marginal role. During AD progression, AChE activity decreases to 55–67% of normal values in the hippocampus and temporal cortex, while the activity of BChE gradually increases in the late stage of AD up to 120% of normal levels and accumulates in senile plaques, thus suggesting an extremely critical role for BChE in ACh hydrolysis and disease progression [3,4]. Therefore, selective BChE inhibitors may also display modulatory activity on Aβ, as found in tacrine and rivastigmine, both lowering Aβ levels in cell cultures [5]. Moreover, dual inhibitors of both AChE and BChE may, indeed, have the potential to interfere with AD along the course of this pathology, as recently reviewed for the clinical applications and limitations in the use of cholinesterase inhibitors for the treatment of AD [6].

Transition metals play a crucial role in vital processes, in particular Cu^2+^, the third bivalent ion after Fe^2+^ and Zn^2+^ in terms of abundance in the human body; all three ions are involved in the regulation of neuronal functions. Studies carried out to assess the relationship between serum copper, zinc, and iron levels with the risk of AD, via a meta-analysis of case-control studies, supported the hypothesis of an association of a high serum copper level to AD onset risk. Serum copper levels were found to be significantly higher in AD patients than in healthy controls, while serum zinc and iron levels were significantly lower in AD patients than in controls [7]. Furthermore, Al^3+^ ions resulting mainly from the intoxication of human exposure were found co-localized with Aβ in senile plaques, but future research is required to recognize its role in AD [8]. 

The role of aluminum (Al^3+^) and zinc (Zn^2+^) is opposite in the onset and progression of Alzheimer’s disease. In fact, several biological assays in an animal model demonstrated that aluminum chloride is able to reduce cognitive capacity in rats [9]. The mechanism involved is yet to be ascertained, but there is evidence that this ion interacts directly with APP metabolism and/or with APOE, a known risk factor in AD [10,11]. Controversially, it has been demonstrated that zinc carboxylate, evaluated towards acetylcholinesterase (AChE) and butyrylcholinesterase (BChE), displayed high inhibitory activity having a strong potency to block cognitive decline in Alzheimer’s disease [12].

Cu^2+^ plays an important role in various physiological processes as a catalytic cofactor for a variety of metal enzymes (superoxide dismutase, cytochrome oxidase, lypoxidase, and tyrosinase). However, it is toxic in biological systems when exceeding cellular requirements by negatively affecting physiological processes, and among them, neurotransmission. Copper dyshomeostasis and accumulation are known to be involved in AD onset and progression [7,13,14,15], and a most rapid accumulation of amyloid plaques in AD patients has been detected [13,14,15]. It has been reported that Cu^2+^ coordination with Aβ modulates Aβ aggregation pattern, potentiating Aβ neurotoxicity and stabilizing smaller soluble Aβ oligomers [16]. The trend for copper-chelating drugs was recently highlighted [17], and a review on metal complexes, including copper complexes with ligands incorporating amyloid-binding moieties for Aβ plaque binding, has been published [18]. Therefore, the development of new probes able to selectively chelate Cu^2+^ remains a challenge towards the discovery of new agents for early AD diagnosis or treatment. 

Recently, the first series of purine nucleosides displaying a promising cholinesterase inhibition has been reported by our group [19].

The nucleoside 6-chloro-9-(2,3,4,6-tetra-*O*-benzyl-α-d-mannopyranosyl)purine derivative (compound **1**, Figure 1) showed a K*i* of ca 2.5 μM for both BChE and AChE, while the analogue 2-acetamidopurine nucleoside exhibiting glycone N^7^ ligation (compound **2**, Figure 1), showed a selective inhibition (K*i* of 50 nM) of BChE. 

Inspired by these first results, we designed a new series of purine nucleosides, aiming to insert a copper chelating molecular moiety for the development of multitarget agents against AD acting as cholinesterase inhibitors and/or as copper chelators. Starting from the general structure in Figure 1, the new compounds were synthesized and assayed as cholinesterase inhibitors aiming to evaluate the role on compound bioactivity of anomeric stereochemistry, sugar d, l configuration, sugar 6-deoxygenation, and glycosyl group regioligation to the purine base at positions N^7^ or N^9^. For preliminary information about protecting group requirements, the peracetylated analogue of the most potent perbenzylated nucleoside was synthesized. New compounds were also tested for their copper chelating efficiency. The results, here presented and discussed, highlight the potent activity of this purine nucleoside family of compounds and demonstrate how copper chelation and anticholinesterase activity/selectivity can be tuned by the regiochemistry of glycone ligation, glycone conformation, and also by the anomeric absolute configuration.

## 2. Results

### 2.1. Chemistry

Synthesis of compounds **5**-**10**, **12, 15** and **16** (Table 1, Figure 1, Figure 2 and Figure 3) was accomplished via TMSOTf catalyzed reaction of fully protected methyl glycosides with the purine base activated by reaction with BSA, a procedure adapted from a methodology previously described by us for other purine nucleosides [19]. The glycosyl donors were the perbenzylated methyl α-d-mannopyranoside (**3**), its 6-deoxy analogue (**4**), the peracetylated 6-deoxy-l-mannose (l-rhamnose) (**11**), and the perbenzylated methyl 6-deoxy-α-l-mannopyranoside (**14**). Reaction conditions optimization was required resulting from the very low yields obtained with the previously reported procedure. Several temperatures and reaction times were tried, under conventional and microwave-assisted syntheses, aiming to improve the overall nucleoside yield. Microwave irradiation at 150 W and 65 °C for 60 min gave the best results, and nucleoside isolated yields are given in Table 1. The four isomers differing in the anomeric configuration and linkage of the glycosyl group to the purine, either at the N^7^ or the N^9^ position, were formed by reacting the perbenzylated methyl α-d-mannopyranoside (**3**), but both the N^9^ anomers were obtained in higher yield than the corresponding N^7^ isomers. However, by reacting the perbenzylated methyl 6-deoxy-d-mannopyranoside donor (**4**), only the N^9^ isomers were obtained in good yield, while the N^7^ isomers were detected as traces (≤1%) by ^1^H NMR. Replacing the benzyl by the acetyl protecting group, the peracetylated l-rhamnose gave the N^9^-linked α-nucleoside (**12**) as the single product, isolated in 63% yield (Table 1, Figure 2). Compound **13** was then obtained in 15% isolated yield by reacting **12** with sodium hydroxide, tetrabutylammonium iodide, and benzyl bromide. Finally, with the perbenzylated methyl 6-deoxy-α-l-mannoside (14) as the starting material, only the N^9^ isomers were obtained, as also observed for the reaction with compound **4** (Table 1, Figure 3).

Both the anomer configuration and the ligation at N^7^ or N^9^ were determined based on ^1^H and ^13^C NMR data and COSY, HMQC, and HMBC experiments. The chemical shift of purine carbon 5 is characteristic and permits to distinguish between the kinetic regioisomer N^7^ and the N^9^ thermodynamic one [19]. These values were between δ 118 and 112 ppm for compounds **7** and **8** (N^7^ isomers), while ligation at N^9^ resulted in chemical shifts of C-5 between ca δ 129 and 122 ppm (as in compounds **5**, **6**, **9**, **10**, **12**, **13**, and **15**, **16**).

The anomeric configuration was assigned from the coupling constant ^3^*J*_1′,2′_ and confirmed by COSY and NOESY experiments. The coupling constants ^3^*J*_1′,2′_ for compounds **5**, **7**, **12,** and **15** were 8.0 Hz, 9.3 Hz, 6.3 Hz, and 7.1 Hz, respectively, thus suggesting the presence of *trans*-diaxial like protons as in ^1^C_4_ conformation for the d series and ^4^C_1_ for the l series. The coupling constant ^3^*J*_1′,2′_ of 7.4 Hz for the β-anomer (**6**) was unexpected, indicating a conformation possibly bearing H-1′ and H-2′ approaching a parallel orientation, tentatively depicted in Figure 1. ^1^H NMR spectra of the β-anomers **8**, **10** and **16** show coupling constants ^3^J_1′,2′_ lower than 1 Hz, as expected. 

By the NMR spectra analysis, the structures of ten nucleosides were proposed, six of them being unexpected. Usually, the most stable glycosyl conformation presents its voluminous protecting groups in equatorial positions, which does not happen in nucleosides **5**, **7**, **9**, **12,** and **15**. Moreover, compound **6** seems to present a non-usual β-conformation, and **12** (α-anomer) was obtained as a single product. These unexpected results led us to perform computational studies, which unambiguously confirmed the proposed structures deduced from NMR data. Figure 2 and Appendix A show the comparison of experimental and calculated NMR data, and the formation of **12** as a single product (Table 2) was unambiguously confirmed. The tables comparing experimental and calculated NMR data can be found in the Appendix A, the latter in line with the experimental results. Interestingly, according to the acquired NMR data and the knowledge to date, all the studied nucleosides exhibited a preferred equatorial orientation of the purine base, which is in full agreement with the exoanomeric effect known for *N*-nucleosides [20].

### 2.2. Anticholinesterase Activity 

The in vitro inhibitory activities of novel purine nucleosides **5**-**10**, **12**, **13** and **15,** and **16** were evaluated against both heterologous (electric eel eeAChE, horse eqBChE) and human (recombinant) cholinesterases isoforms, by applying the Ellmann colorimetric assay, with slight modifications [21,22]. The results are summarized in Table 3, thus reporting the half maximal inhibitory concentration (IC_50_) values, or the percent of inhibition at the final tested concentration of 20 μM, with an exception for compounds **15** and **16**, which were tested at 5 μM due to their lower water solubility, but they were not able to produce more than 50% of inhibition for eeAChE and eqBChE. Compounds **6, 8**, **9**, **10,** and **13** were the most interesting ones. Nucleosides **8** and **9** show eeAChE IC_50_ values of 2.6 and 4.7 μM, **6** and **13** show eqBChE IC_50_ values of 4.3 and 5.1 μM, respectively, whilst compound **10** proved moderate inhibition of BChE. Furthermore, compound **9** was an eeAChE selective inhibitor, while compounds **7** (IC_50_ = 10 μM) and **13** were selective for eqBChE. The most potent compound was the dual inhibitor compound **8**.

The reported activities highlight that sugar, the geometry of the glycosidic bond, and the regioisomerism of substitution pattern on the nucleoside affected the anticholinesterase activity. It has been, indeed, reported that depending on sugars stereochemistry, the acceptor/donor hydrogen bond properties of glycosylated compounds can significantly change and deeply affect binding properties to the target [23].

The relationship between glycone d-configuration, N-ligation, and selectivity is in agreement with our previous findings for related nucleosides with a 6-chloropurine in their structure, while those for the l-configuration are herein reported for the first time [19,24].

**Table 3 pharmaceuticals-16-00054-t003:** Half maximal inhibitory concentration (IC_50_) values of AChE and BChE inhibition by nucleosides **1**–**2**, **5**–**10**, **12**–**13**, **15**–**16**.

Compound nr.		IC_50_ (μM) ^a^		
	eeAChE ^b^	hAChE ^c^	eqBChE ^d^	hBChE ^c^
**1**	2.40 ± 0.30	-	2.80 ± 0.30	-
**2**	18.10 ± 4.80	-	0.05 ± 0.01	-
**5**	>20 [25 ± 5]	-	>20 [11 ± 7]	-
**6**	10.60 ± 1.10	17.00 ± 0.75	4.29 ± 1.70	4.58 ± 0.52
**7**	>20	-	10.00 ± 1.10	-
**8**	2.61 ± 1.23	4.81 ± 1.24	4.40 ± 1.30	4.55 ± 1.05
**9**	4.69 ± 0.51	>20 [25 ± 6]	>20 [28 ± 3]	15.4 ± 1.10
**10**	15.0 ± 2.1	17.8 ± 2.20	11.7 ± 1.2	8.70 ± 1.35
**12**	>20 [8 ± 5]	-	>20 [5 ± 3]	-
**13**	>20 [6 ± 3]	>20 [8 ± 5]	5.12 ± 0.25	>20 [25 ± 4]
**15**	>5 [31 ± 4]	>5 [28 ± 3]	>5 [24 ± 6]	>5 [18 ± 5]
**16**	>5 [18 ± 3]	>5 [16 ± 9]	>5 [26 ± 6]	>5 [30 ± 7]
Donepezil	0.021± 0.020	0.016 ± 0.002	2.75 ± 0.20	4.80 ± 0.05
Galantamine	0.56 ± 0.12	5.000 ± 0.071 ^e^	12.0 ± 0.3	59.2 ± 1.7 ^e^

^a^ IC_50_ values determined by interpolation of the sigmoidal dose-response curves as obtained by regression with GraphPad Prism software (version 5.01) of at least seven different data points, or inhibition percentage at the higher tested concentration (20 μM) in parentheses; data are means ± SD of three independent measurements. Donepezil and Galantamine were used as positive controls. Reported data have been experimentally obtained, and values found accordingly with the literature. ^b^ Electric eel acetylcholinesterase. ^c^ Recombinant human isoform. ^d^ Horse serum butyrylcholinesterase. ^e^ Values obtained from the literature determined for brain AChE and serum BChE [25].

With the exception of the 2,3,4-tri-*O*-benzyl-6-deoxy-α-d-mannosyl purine derivative **9**, exhibiting a glycone ^1^C_4_ conformation and reported as a selective AChE inhibitor, compounds bearing a 2,3,4,6-tetra-*O*-benzyl-β-d-mannosyl fragment (**6** and **8**) showed from moderate to good inhibition of heterologous AChE and BChE and resulted most active than the quite inactive corresponding α anomers. Moreover, regarding N^9^ or N^7^ ligation of sugar, the N^7-^linked β-glycosyl regioisomer **8** proved a better activity than the N^9^ regioisomers against both cholinesterases, thus resulting in compound **8** as a dual inhibitor against human cholinesterase isoforms as well.

Amongst the ten synthesized nucleosides, the peracetylated N^9^-linked 6-deoxy-α-l-*manno* nucleoside **12** was not active towards any of the cholinesterases, in agreement with previous findings, which revealed that aromatic rings are preferred for the anticholinesterase activity of the studied purine nucleosides [24], and all the benzylated compounds, except the *N*^9^-linked α-d-*manno* nucleoside **5**, were active at some extent. However, benzylation of amide nitrogen in N^9^-linked *N*-benzyl 6-deoxy-α-l-*manno* nucleoside **13**, exhibiting a glycosyl ^4^C_1_ conformation, deeply modified the activity profile, thus inducing a selective inhibition of the equine BChE, which disappeared in human BChE. 

Substitution of d-mannosyl with l-mannosyl sugar moiety resulted in a dramatic reduction of aqueous solubility in compounds **15** and **16**, which were tested at a lower concentration (5 μM) without showing any significant inhibition.

In order to verify compound behavior against the human cholinesterases, the most potent inhibitors (**6**, **8,** and **9**), dual inhibitor **10,** and nucleosides **15** and **16** were tested against hAChE and hBChE. The previously known dual inhibitors **6**, **8,** and **10** seemed to keep this property, however, with a slight IC_50_ value variation. Compound **6** IC_50_ value for hAChE increased from 10.6 to 17.0 μM, while the value for hBChE did not undergo such a variation (from 4.3 to 4.6 μM), suggesting a higher affinity of this compound to BChE. Nucleoside **8**, with IC_50_ values of 4.8 μM and 4.6 μM for hAChE and hBChE, respectively, seemed to keep the dual inhibition with no greater affinity for one of the enzymes. Furthermore, for compound **10**, a higher IC_50_ value for hAChE (IC_50_ = 17.8 μM) was found when compared to eeAChE. However, the IC_50_ value found for hBChE was lower than the one found for eqBChE. Nucleoside **9**, a selective eeAChE inhibitor, in the human enzymes inhibitory assays turned into a hBChE selective inhibitor, while compound **13** showed no activity against the human enzymes. For compounds **15** and **16**, the lower concentration which was tested was not able to produce 50% of enzyme inhibition. Finally, also in Table 3, the inhibition results for the latter compounds at 5 μM final concentration are disclosed. Nucleosides **15** seemed to have a slightly higher affinity to AChE, producing 31 and 28% inhibition of eeAChE and hAChE, respectively, while **16** seemed to present a higher inhibition towards BChE, with a higher inhibition value for the hBChE (30%). However, both seem to show potential as non-selective inhibitors. These results then suggest that by improving these compounds’ water solubility, the inhibition rates could increase, and the nucleosides could become interesting candidates as cholinesterase inhibitors.

### 2.3. Chelating Activity 

The chelating activity of all purine nucleoside probes was evaluated by the UV-Visible method in order to determine compounds’ absorption spectra in the absence and in the presence of Cu^2+^ at concentrations ranging from 1 μM to 10 μM. Amongst nucleosides **5**-**10** and **12**, the latter displayed the most interesting profile (Figure 3). The resulting wavelength shift from λ = 280 to λ = 325 nm indicates a conformational rearrangement induced by interaction with Cu^2+^. Furthermore, selectivity towards the biologically relevant ions Zn^2+^, Fe^2+,^ and Al^3+^ was also investigated, thus demonstrating for most of them a negligible quenching. 

Hence, although **12**, the only acetylated nucleoside, did not show promising cholinesterase inhibition, it became the first nucleoside-based molecule with metal chelating properties and, therefore, became a starting point for the investigation of structural modifications that could lead to the desired dual-target compounds against AD.

In order to understand which structural alterations could be carried out to achieve our goal, computational studies were performed. Molecular electrostatic potential maps, in which the red and blue hues represent negatively and positively charged regions, respectively, and green tones depict neutral regions, were generated for compounds **5**-**7**, **9**, and **12**. In all cases, a significant electron density in the amide group of the purine base was found, and the benzyl units were found to be almost neutral. Moreover, compound **7**, exhibiting N^7^ purine ligation, also exhibits high electron density between atoms N3 and N9, and compound **12** shows some negative density in the acetyl groups. Hereinafter, the generated maps for compounds **7** and **12** are disclosed (Figure 4), the remaining being found in the Appendix A).

Based on the molecular electrostatic potential map generated for compound **12**, the geometry for the complex compound **12**-Cu^2+^ was optimized, starting with the copper ion placed near the amide group of the purine. The electronic transitions in water for the two possible conformers, “b” and “c,” of compound **12** in the absence and in the presence of Cu^2+^ ions are presented in Figure 5. In comparison with the experimental results, the calculated transitions are displaced to shorter wavelengths; however, the bathochromic shift observed experimentally with the addition of Cu^2+^ is also noticed in the calculated results. The larger red shift in λ_max_ from 254 to 284 nm is observed for conformer “b.” Also, the energy difference between both complexes listed in Table 4 shows that complex **12**a-Cu^2+^ is more stable. Finally, the optimized molecular geometry of complexes **12**-Cu^2+^ is shown in Figure 6, and the electrostatic potential map for the most stable complex, the **12**a-Cu^2+^ complex, is presented in Figure 7.

Furthermore, having noticed that the glycosyl groups have a small contribution to the molecule electrostatic potential, the interaction with the metal ions of the purine in conformer “b” was studied after replacing the glycosyl unit of compound **12** with a hydrogen atom at N9. In Figure 8, the simulated absorption spectra based on electronic transitions calculated for 6-benzamidopurine in the absence and in the presence of the metal ions in water are presented. The spectra were simulated based on the assumption that all the solutions have the same concentration (10 μM) and that the metal ions are totally binded to the purine. As observed experimentally, the calculated absorption spectra indicate that the binding constant between the purine and Cu^2+^ is larger than the one with the other metal ions. Also, it is important to stress that the spin multiplicity of Cu^2+^ is doublet while all other metal ions’ spin multiplicity is singlet.

Having the computational studies performed, three new nucleosides, **13**, **15,** and **16**, were synthesized aiming at confirming the latter findings, **13** being a benzylated analogue of compound **12** with an additional *N*-benzyl group, **15** a benzylated analogue of **12** and **16** the β-anomer of **15**. Amongst these new compounds, **15** and **16** displayed the most interesting profiles since **13** did not show chelating activity towards any of the biologically relevant ions. Compound **15** was found to be a selective copper chelator, as it is shown in Figure 9. In the complex **15**-Cu^2+^ spectrum, there is a perceptible wavelength shift from λ = 284 to λ = 334 nm, indicating a conformational rearrangement due to the interaction with Cu(II). Through the analysis of the remaining spectra, it is possible to verify that there is no relevant interaction of this nucleoside with the other ions. Compound **16** was shown to be a non-selective chelator, as the resulting spectra showed conformational rearrangements induced by interaction with all the tested ions, as can be verified in Figure 10. Interactions with copper, zinc, iron and aluminum ions result in wavelength shifts from λ = 290 nm to λ = 345, 346, 339, and 351 nm, respectively.

To conclude, the computational studies revealed that for metal chelation, the glycosyl units had a small contribution, being the amide group of the purine the most relevant. The results obtained with compounds **13** and **15** confirmed these findings. Nucleoside **15**, the benzylated analogue of **12**, also showed selective copper chelation over the remaining metals, proving that the glycosyl-protecting groups had no influence on this property. Moreover, compound **13**, being similar to **15** with an extra *N*-benzyl group in the purine moiety, showed no activity, showing that the metal chelation site should be found in the purine. Finally, the computational studies indicated that this purine base should have a higher affinity to Cu^2+^; however, experimentally, compound **16** showed no selectivity over the four metal ions.

## 3. Discussion

A series of novel nucleosides were synthesized and tested, leading to new insights into the structural features required for BChE or AChE selective inhibition and Cu^2+^ binding. Starting from lead compounds 1 and 2, the replacement of 6-chloropurine by 6-benzamidopurine led to two novel and selective Cu^2+^ chelating agents 12 (with the peracetylated α-l-rhamnosyl group) and its analogue perbenzylated compound 15 (with the perbenzylated α-l-rhamnosyl group). Moreover, the latter modifications led to a novel metal chelator 16 (^1^C_4_ conformer), obtained with the same substrate as 15, with an affinity for Cu^2+^, Zn^2+^, Fe^2+^, and Al^3+^. Importantly, all synthesized nucleosides showed the equatorial orientation of the base, most probably resulting from the exoanomeric and steric effects. Amongst the new nucleosides tested, 12, the only acetylated compound, did not show cholinesterase inhibition, while all the benzylated compounds, except 5 (the N^9-^linked α-d-mannosyl nucleoside), were active to some extent, highlighting the importance of aromatic rings for this property. With regard to eeAChE and eqBChE, the selective inhibitors exhibit a glycosyl ^1^C_4_ conformation regardless of the ligation position to the purine base, while the dual inhibitors show a glycone ^4^C_1_ conformation or derived thereof. Moreover, N^9^ ligation is shown for AChE and N^7^ ligation for BChE, in agreement with our previous results shown by the 6-chloropurine series [19]. However, for human cholinesterases, only compound 9 (N^9^-linked 6-deoxy-α-d-*manno*) was selective for BChE inhibition, and no nucleoside showed AChE selective inhibition, while the known eeAChE and eqBChE dual inhibitors kept this property, with 6 (N^9-^linked β-d-*manno*) and 10 (N^9^-linked 6-deoxy-β-d-*manno*) having a higher affinity for BChE and 8 (N^7^-linked β-d-*manno*) showing no greater affinity for any of the enzymes. Finally, compounds 15 and 16, with the glycone l configuration, N^9^ purine ligation, and aromatic rings in their structure, showed metal chelation and some inhibition of the cholinesterases, both the animal and the human ones, at the tested concentration, becoming the first nucleoside-based molecules with potential to act as dual-target compounds against AD. This work illustrates the role of sugars by tuning bioactivity and selectivity, encouraging further investigation of this family of purine nucleosides towards glycone structure optimization to access anticholinesterase compounds and copper chelators as multitarget agents against AD. 

## 4. Materials and Methods

Reagents were purchased from commercial suppliers (Sigma Aldrich, St. Louis, MI, USA) and used without further purification. The starting materials **3**, **4,** and **11** were available from Sigma Aldrich, Glentham Life Sciences (Corsham, UK), Toronto Research Chemicals (North York, ON, Canada). Compound **14** was obtained by benzylation of methyl α-d-rhamnoside purchased from Alfa Aesar. The physical and spectroscopic data of 14 was in full agreement with those previously reported [26]. Microwave-assisted synthesis was performed with a CEM Discover and Explorer SP. NMR spectra were recorded on a BRUKER Advance 400 spectrometer at 298 K or on a BRUKER Advance III 500 spectrometer at 298 K. Spectra were referenced internally to residual proton-solvent (^1^H) or carbon-solvent (^13^C) resonances, and reported relative to tetramethylsilane (0 ppm). Chemical shifts (δ) are given in ppm and coupling constants (J) in Hz. UV spectra were recorded on a SHIMADZU UV-1800 spectrometer with quartz cells. Optical rotations were determined on a Perkin-Elmer 341 polarimeter and on an Anton Paar MCP 100 polarimeter. HRMS analyses were performed on an Agilent 6530 Accurate Mass Q-TOF, only significant m/z peaks, with their percentage of relative intensity in parentheses, are reported. TLC was performed on silica gel (Merck 5554), and column chromatography for compound isolation was carried out with silica gel. Solvents were dried before use according to the conventional procedures. 

### 4.1. Synthesis

*N*-glycosylation procedure: *N*,*O*-Bis(trismethylsilyl)acetamide (BSA) (3.0 equiv.) was added to the solution of *N*-(9*H*-purin-6-yl)benzamide (1.5 equiv.) in dry acetonitrile. The mixture was stirred at room temperature for 50 min. Then the fully protected methyl mannopyranoside (**3**, **4**, **11,** and **14**) (1 mmol) was dissolved in dry acetonitrile, and trimethylsilyl trifluoromethanesulfonate (TMSOTf) (8 equiv.) was added to the reaction mixture. The reaction was carried out under microwave irradiation (150 W, 65 °C, 60 min). The mixture was poured into dichloromethane, washed with a saturated solution of NaCl, and extracted with dichloromethane (4 × 25 mL). The combined organic phases were washed with brine, dried (MgSO_4_), and concentrated. Compounds were isolated and purified by column chromatography (cyclohexane/EtOAc from 9:1 to 6:4).

Compound **12** benzylation procedure: Powdered sodium hydroxide (15.0 equiv.) and tetrabutylammonium iodide (TBAI) (0.75 equiv.) were added to a solution of **12** in THF (20 mL/mmol). The mixture was stirred for 10 min at room temperature. Then, benzyl bromide [12.0 equiv. (3.0 equiv./OH group)] was added, and the mixture was stirred for 3 h. The mixture was poured into dichloromethane, washed with water, and extracted with dichloromethane (3 × 100 mL). The combined organic phases were dried (MgSO_4_), filtered, and concentrated. Compound **13** was isolated and purified by column chromatography (hexane/EtOAc from 10:0 to 6:1).

6-benzamido-9-(2,3,4,6-tetra-*O*-benzyl-α-d-mannopyranosyl)purine (**5**), 6-benzamido-9-(2,3,4,6-tetra-*O*-benzyl-β-d-mannopyranosyl)purine (**6**), 6-benzamido-7-(2,3,4,6-tetra-*O*-benzyl-α-d-mannopyranosyl)purine (**7**) and 6-benzamido-7-(2,3,4,6-tetra-*O*-benzyl-β-d-mannopyranosyl)purine (**8**).

Data for compound **5**, using compound **3** as starting material: Syrup. Yield = 22%. Rf = 0.85 (EtOAc/Hexane 4:1). [α]_D_^20^ = −24° (*c* 2, MeOH). ^1^H NMR (400 MHz, CD_3_OD): δ 8.58 (s, 1H, H-2), 8.42 (s, 1H, H-8), 8.09 (d, 2H *ortho*, J = 7.0 Hz, PhCO), 7.65 (t, 1H *para*, J = 7.0 Hz, PhCO), 7.57 (t, 2H *meta*, J = 7.0 Hz, PhCO), 7.36–6.89 (m, 21H, Ph-Bn, NH), 6.13 (d, 1H, H-1′, J_1′,2′_ = 8.0 Hz), 4.78 (dd, 1H, H-2′, J_2′,3′_ = 2.8 Hz, superimposed to methanol contamination of the solvent), 4.71, 4.68, 4.66, 4.63 (AB system, 2H, Bn-CH_2_, J = 11.7 Hz), 4.60-4.43 (m, 8H, H-5′, 3xAB system, 3x CH_2_Ph_,_ part A of AB system, CH_2_Ph), 4.25, 4.22 (AB system, 1H, CH_2_Ph, J = 12.0 Hz), 4.11 (t, 1H, H-3′, J_3′,4′_ = 2.6 Hz), 3.85 (dd, 1H, H-4′, J_4′,5′_ = 5.5 Hz), 3.76 (dd, 1H, H-6′a, J_6′a,5′_ = 6.7 Hz, J_6′a,6′b_ = 10.9 Hz), 3.65 (dd, 1H, H-6′b, J_6′b,5′_ = 3.9 Hz). ^13^C NMR (100 MHz, CD_3_OD): δ 167.4 (CONH), 156.6 (C-4), 153.4 (C-2), 150.5 (C-6), 145.6 (C-8), 139.7(2) (Ph-Cq), 139.7 (Ph-Cq), 139.6 (Ph-Cq), 138.7 (Ph-Cq), 134.4 (Ph-Cq), 134.3 (Ph-Cq), 130.1, 129.9, 129.8, 129.7(3), 129.6(7), 129.6(0), 129.5(6), 129.4(3), 129.3(5), 129.3(1), 129.2(6), 129.2, 129.0, 127.9 (C-5), 81.9 (C-1′), 77.7 (CH_2_Ph), 77.5 (CH_2_Ph), 76.7 (C-4′), 75.8 (C-3′), 74.6 (CH_2_Ph), 74.4 (C-2′), 74.1, 73.4, 72.7 (C-5′, 2x CH_2_Ph), 70.2 (C-6′). HRMS (CI^+^); calcd for C_46_H_43_N_5_O_6_ (M+Na)^+^: 784.3106, found: 784.3162.

Data for compound **6** using compound **3** as starting material: Syrup. Yield = 16%. Rf = 0.70 (EtOAc/Hexane 3:2). [α]_D_^20^ = −6° (*c* 2, MeOH). ^1^H NMR (400 MHz, CDCl_3_): δ 8.77 (s, 1H, H-2), 8.09 (s, 1H, H-8), 8.08 (d, 2H *ortho*, J = 6.0 Hz, PhCO), 7.63 (t, 1H *para*, J = 7.0 Hz, PhCO), 7.55 (t, 2H *meta*, J = 7.0 Hz, PhCO), 7.37–7.01 (m, 21H, Ph, NH), 6.08 (d, 1H, H-1′, J_1′,2′_ = 7.4 Hz), 4.97 (dd, 1H, H-2′, J_2′,3′_ = 2.9 Hz), 4.76, 4.73 (part A of AB system, 1H, CH_2_Ph, J = 12.0 Hz), 4.68, 4.65 (part B of AB system, 1H, CH_2_Ph, J = 12.0 Hz), 4.55–4.43 (m, 6H, H-5′, 2xAB system, CH_2_Ph, part A of AB system, CH_2_Ph), 4.30, 4.27 (part B of AB system, 1H, CH_2_Ph, J = 12.0 Hz), 3.99 (dd, 1H, H-3′, J_3′,4′_ = 3.4 Hz), 3.91 (dd, 1H, H-4′, J_4′,5′_ = 6.0 Hz), 3.78 (dd, 1H, H-6′a, J_5′,6′a_ = 6.0 Hz, J_6′a,6′b_ = 12.8 Hz), 3.71 (d, 1H, H-6′b, J_6′b,5′_ = 3.8 Hz). ^13^C NMR (100 MHz, CDCl_3_): δ 164.6 (CONH), 152.6 (C-2), 151.6 (C-4), 149.6 (C-6), 142.8 (C-8), 137.9 (Ph-Cq), 137.9 (Ph-Cq), 137.7 (Ph-Cq), 137.5 (Ph-Cq), 136.8 (Ph-Cq), 136.8 (Ph-Cq), 128.9, 128.5, 128.4, 128.3, 128.0(0), 127.9(8), 127.9(5), 127.8 (Ph), 123.2 (C-5), 81.1 (C-1′), 75.7 (C-5′), 74.9, 74.8 (C-3′, C-4′), 73.2 (CH_2_Ph, C-2′), 72.4 (CH_2_Ph), 72.4 (CH_2_Ph), 72.1 (CH_2_Ph), 68.4 (C-6′). HRMS (CI^+^); calcd for C_46_H_43_N_5_O_6_ (M+H)^+^: 656.2867, found: 656.2856.

Data for compound **7** using compound **3** as starting material: Syrup. Yield = 6%. Rf = 0.40 (EtOAc/MeOH 4:1). [α]_D_^20^ = + 5° (*c* 3, MeOH). ^1^H NMR (400 MHz, CD_3_OD): δ 8.25 (s, 1H, H-2), 8.14 (s, 1H, H-8), 7.35-6.60 (m, 26H, Ph, NH), 5.84 (d, 1H, H-1′, J_1′,2′_ = 9.3 Hz), 4.66-4.40 (m, 9 H, H-3′, 4xAB system, CH_2_Ph), 4.12-4.06 (m, 2H,H-4′, H-6′a), 3.91 (m, 1H, H-5′), 3.85 (d, 1H, H-2′), 3.75 (d, 1H, H-6′b, J_6′a,6′b_ = 10.8 Hz); ^13^C NMR (100 MHz, CD_3_OD): δ 169.1 (CONH), 157.3 (C-4), 153.3, 153.2 (C-2, C-6),146.7 (C-8), 139.7, 139.4, 139.1, 137.8, 136.1, 134.5, 134.4, 130.7, 130.1, 130.0, 129.9, 129.8, 129.7, 129.6(6), 129.6(2), 129.5(5), 129.4, 129.3, 116.2 (C-5), 81.6 (C-1′), 80.3 (C-3′), 75.4 (CH_2_Ph), 75.0 (CH_2_Ph), 74.4 (C-2′), 74.3 (CH_2_Ph), 73.8 (CH_2_Ph), 73.7 (C-4′), 71.8 (C-5′), 67.9 (C-6′). HRMS (CI^+^); calcd for C_46_H_43_N_5_O_6_ (M+H)^+^: 762.3286, found: 762.3293.

Data for compound **8** using compound **3** as starting material: Syrup. Yield = 12%. Rf = 0.20 (EtOAc/MeOH 4:1). [α]_D_^20^ = −5° (*c* 2.7, MeOH). ^1^H NMR (400 MHz, CD_3_OD): δ 8.31 (s, 1H, H-8), 8.16 (s, 1H, H-2), 7.43–7.06 (m, 24H, Ph, NH), 6.78 (d, 2H, Ph, J = 7.59 Hz), 6.08 (s, 1H, H-1′), 4.91–4.75 (m, 3H, AB system CH_2_Ph-3′ + part A of AB system CH_2_Ph-4′, superimposed by methanol contamination of the solvent), 4.65, 4.62 (part A of AB system, J = 11.0 Hz, 1H, CH_2_Ph-2′), 4.64, 4.61 (part A of AB system, J = 11.0 Hz, 1H, CH_2_Ph-6′), 4.58, 4.55 (part B of AB system, J = 10.0 Hz, 1H, CH_2_Ph-4′), 4.46, 4.43 (part B of AB system, J = 11.0 Hz, 1H, CH_2_Ph-6′), 4.31 (t, 1H, H-4′, _J4′,3′_= 9.7 Hz, J_4′,5′_ = 9.6 Hz), 4.21 (br s, 1H, H-2′), 4.14, 4.12 (part B of AB system, J = 11.0 Hz, 1H, CH_2_Ph-2′), 4.00 (dd, 1H, H-3′, J_3′,2′_= 2.3 Hz), 3.84 (d, 1H, H-6′a, J_6′a,6′b_ = 10.4 Hz), 3.78 (d, 1H, H-5′), 3.72 (dd, 1H, H-6′b). ^13^C NMR (100 MHz, CD_3_OD): δ 176.5 (CONH), 159.8 (C-4), 153.7 (C-6), 153.2 (C-2), 145.0 (C-8), 139.6(2) (Ph-Cq), 139.5(9) (Ph-Cq), 139.1 (Ph-Cq), 138.2 (Ph-Cq), 129.6, 129.4(0), 129.3(6), 129.3, 129.2, 129.0 (Ph), 113.6 (C-5), 86.9 (C-1′), 83.9 (C-3′), 79.1 (C-5′), 78.0 (C-2′), 76.3 (CH_2_Ph), 76.2 (CH_2_Ph), 74.3 (C-4′), 74.2 (CH_2_Ph), 73.7 (CH_2_Ph), 68.4 (C-6′). HRMS (CI^+^); calcd for C_46_H_43_N_5_O_6_ (M+H)^+^: 762.3286, found: 762.3276.

6-benzamido-9-(6-deoxy-2,3,4-tri-*O*-benzyl-α-d-mannopyranosyl)purine (**9**), 6-benzamido-9-(6-deoxy-2,3,4-tri-*O*-benzyl-β-d-mannopyranosyl)purine (**10**)

Data for compound **9** using **4** as starting material: Syrup. Yield = 33%. Rf = 0.8 (DCM/EtOAc 1:1). [α]_D_^20^ = +18 (*c* 1, MeOH). ^1^H NMR (400 MHz, CD_3_OD): δ 8.58 (s, 1H, H-2), 8.42 (s, 1H, H-8), 8.08 (d, 2H, *ortho*, J = 7.3 Hz, PhCO), 7.64 (t, 1H, *para* J = 7.6 Hz, PhCO), 7.55 (t, 2H, *meta*, J = 7.6 Hz, PhCO), 7.40–7.27 (m, 11H, Ph, NH), 7.12–7.05 (m, 3H, Ph), 6.91 (d, 2H, Ph), 6.15 (d, 1H, H-1′, J_1′,2′_ = 8.1 Hz), 4.79 (dd, 1H, H-2′, J_2′,3′_ = 2.7 Hz), 4.75, 4.72, 4.70, 4.67 (AB system, 2H, OCH_2_-3′, J = 12.1 Hz), 4.59, 4.56, 4.56, 4.53 (AB system, 2H, OCH_2_-4′), 4.45, 4.42 (part A of AB system, 1H, OCH_2_-2′, J = 12.3 Hz), 4.37 (t, 1H, H-5′, J_5′,4′_ = 5.6 Hz, J_5′,6′_ = 6.54 Hz), 4.25, 4.22 (part B of AB system, 1H, OCH_2_-2′, J = 12.3 Hz), 4.10 (t, 1H, H-3′, J_3′,4′_ = J_3′,2′_ = 2.88 Hz), 3.62 (dd, 1H, H-4′), 1.34 (d, 3H, CH_3_, J = 6.54 Hz). ^13^C NMR (100 MHz, CD_3_OD): δ 168.4 (CO), 153.7 (C-4), 153.4 (C-2), 153.4 (C-6), 145.5 (C-8), 139.8, 139.7, 138.8, 134.2, 130.1, 129.8(2), 129.8(1), 129.7, 129.6(2), 129.5(7), 129.5, 129.4, 129.2(9), 129.2(5) (Ph), 126.4 (C-5), 81.6 (C-1′), 81.4 (C-4′), 76.2 (C-3′), 74.9 (C-2′), 74.4 (C-5′), 74.1 (OCH_2_-3′), 73.6 (OCH_2_-4′), 72.7 (OCH_2_-2′), 18.6 (CH_3_). HRMS (CI+); calcd for C_39_H_37_N_5_O_5_ (M+H)+: 656.2867, found: 656.2856.

Data for compound **10** using **4** as starting material: Syrup. Yield = 40%. Rf = 0.7 (DCM/EtOAc 1:1). [α]_D_^20^ = +4 (*c* 1, MeOH). ^1^H NMR (400 MHz, CD_3_OD): δ 8.54 (s, 1H, H-2), 8.32 (s, 1H, H-8), 8.08 (d, 2H, *ortho*, J = 7.4 Hz, PhCO), 7.64 (t, 1H, *para*, J = 7.5 Hz, PhCO), 7.55 (t, 2H, *meta*, J = 7.8 Hz, PhCO), 7.41 (d, 2H, J = 6.9 Hz, Ph), 7.34–7.25 (m, 9H, NH, Ph), 7.01–6.99 (m, 3H, Ph), 6.87–6.84 (m, 2H, Ph), 5.94 (s, 1H, H-1′), 4.95, 4.92 (part A of AB system, 1H, OCH_2_-4′, J = 11.3 Hz), 4.86, 4.83 (part A of AB system, 1H, OCH_2_-3′, J = 12 Hz), 4.79-4.64 (m, 3H, part B of AB system, OCH_2_-3′, part B of AB system, OCH_2_-4′, part A of AB system, OCH_2_-2′), 4.30, 4.28 (part B of AB system, 1H, OCH_2_-2′, J = 11.58 Hz), 4.23 (br s, 1H, H-2′), 3.97 (dd, 1H, H-3′, J_3′,2′_ = 1.4 Hz, J_3′,4′_ = 7.1 Hz), 3.74–3.64 (m, 2H, H-4′,H-5′), 1.45 (d, J_5′,6′_ = 5.8 Hz, 3H, CH_3_). ^13^C NMR (100 MHz, CD_3_OD): δ 168.0 (CO), 152.9 (C-2), 151.8 (C-6), 150.9 (C-4), 144.1 (C-8), 139.8 (PhCq), 139.6 (PhCq), 138.1 (PhCq), 133.9 (PhCq), 129.8, 129.6(2), 129.5(6), 129.4(4), 129.3(5), 129.2, 129.1, 129.0, 128.8(9), 128.7(8) (Ph), 124.3 (C-5), 84.0 (C-3′), 83.2 (C-1′), 80.6 (C-4′), 76.4 (C-5′), 76.3 (OCH_2_-2′), 75.5 (OCH_2_-4′), 74.5 (C-2′), 73.8 (OCH_2_-3′), 18.4 (C-6′). HRMS (CI^+^); calcd for C_39_H_37_N_5_O_5_ (M+H)^+^: 656.2867, found: 656.2870. 

6-benzamido-9-(6-deoxy-2,3,4-tri-*O*-acetyl-α-l-mannopyranosyl)purine (**12**), 6-benzamido-*N*-benzyl*-9-(*6-deoxy-2,3,4-tri-*O*-benzyl-α-l-mannopyranosyl)purine(**13**)

Data for compound **12** using **11** as starting material: Syrup. Yield = 63%. Rf = 0.7 (DCM/EtOAc 1:1). [α]_D_^20^ = −4 (*c* 1, MeOH). ^1^H NMR (400 MHz, CD_3_OD): δ 8.75 (s, 1H, H-2), 8.59 (s, 1H, H-8), 8.07 (d, 2H, *ortho*, J = 7.2 Hz, PhCO), 7.65 (t, 1H, *para*, J = 7.5 Hz, PhCO), 7.57–7.54 (m, 3H, *meta* PhCO, NH), 6.30 (d, 1H, H-1′, J_1′,2′_ = 6.3 Hz), 6.26 (dd, 1H, H-2′, J_2′,3′_ = 3.5 Hz), 5.65 (dd, 1H, H-3′, J_3′,4′_ = 5.6 Hz), 5.01 (t, 1H, H-4′, J_4′,5′_ = 5.6 Hz,), 4.28 (qd, 1H, H-5′, J_5′,6′_ = 5.2 Hz), 2.14 (s, 3H, CH_3_), 2.12 (s, 3H, CH_3_), 1.94 (s, 3H, CH_3_), 1.41 (d, 3H, CH_3_). ^13^C NMR (100 MHz, CD_3_OD): δ 171.4(5) (CO), 171.3 (CO), 171.0 (CO), 168.3 (COPh), 153.5 (C-4, C-2), 151.3 (C-6),144.7 (C-8), 134.9, 134.0 (Cq, Ph), 129.8 (Ph), 129.5 (Ph), 125.0 (C-5), 123.3 (Ph), 120.2 (Ph), 80.3 (C-1′), 73.2 (C-4′), 73.1 (C-5′), 70.3 (C-3′), 68.9 (C-2′), 20.7 (CH_3_), 20.6 (CH_3_), 20.4 (CH_3_), 17.2 (CH_3_). HRMS (CI^+^); calcd for C_24_H_25_N_5_O_8_ (M+H)^+^: 512.1776, found: 512.1775.

Data for compound **13** using **12** as starting material: Syrup. Yield = 15%. Rf = 0.8 (Hex/EtOAc 2:1). [α]_D_^20^ = −90 (*c* 0.1, MeOH). ^1^H NMR (500 MHz, CD_3_OD): δ8.53 (s, 1H, H-2), 8.28 (s, 1H, H-8), 7.48-7.01 (m, 25H, Ph), 6.08 (d, 1H, H-1′, J_1′,2′_= 8.26 Hz), 5.57 (s, 2H, NCH_2_Ph,), 4.69, 4.67, 4.65, 4.63 (AB system, 2H, OCH_2_-3′, J = 12.05 Hz), 4.64 (dd, 1H, H-2′, J_2′,3′_ = 2.81 Hz), 4.52 (s, 2H, OCH_2_-4′), 4.23 (qd, 1H, H-5′, J_4′,5′_ = 5.23 Hz, J_5′,6′_ = 6.70 Hz), 4.16, 4.14, 4.07, 4.05 (AB system, 2H, OCH_2_-2′, J = 11.98 Hz), 4.00 (br t, 1H, H-3′, J_3′,4′_=3.34 Hz), 3.59 (dd, 1H, H-4′), 1.33 (d, 1H, H-6′). ^13^C NMR (100 MHz, CD_3_OD): δ 174.1 (CO), 155.3 (C-6), 154.5 (C-4), 153.3 (C-2), 146.2 (C-8), 139.8, 139.7, 138.8, 138.7 (CqPh), 137.6, 132.4, 130.1, 129.8, 129.7(7), 129.7, 129.6, 129.5(9), 129.4, 129.3, 129.2, 128.9 (Ph), 128.7 (C-5), 80.9 (C-4′), 80.8 (C-1′), 76.1 (C-3′), 74.8 (C-2′), 74.6 (C-5′), 74.0 (OCH_2_-3′), 73.5 (OCH_2_-4′), 72.7 (OCH_2_-2′), 52.7 (NCH_2_Ph), 18.3 (C-6′). HRMS (CI^+^); calcd for C_39_H_38_N_5_O_5_ (M+H)_+_: 746.3354, found: 746.3337.

6-benzamido-9-(6-deoxy-2,3,4-tri-*O*-benzyl-α-l-mannopyranosyl)purine (**15**), 6-benzamido-9-(6-deoxy-2,3,4-tri-*O*-benzyl-β-l-mannopyranosyl)purine (**16**)

Data for compound **15** using **14** as starting material: Syrup. Yield = 8%. Rf = 0.9 (DCM/EtOAc 1:1). [α]_D_^20^ = −75 (*c* 0.2, MeOH). ^1^H NMR (400 MHz, CD_3_OD): δ 8.59 (s, 1H, H-2), 8.41 (s, 1H, H-8), 8.09 (d, 2H, *ortho*, J = 7.38 Hz, PhCO), 7.63 (t, 1H, *para*, J = 6.98 Hz, PhCO), 7.54 (t, 2H, *meta*, J = 7.86 Hz, PhCO), 7.41-7.24 (m, 11H, Ph, NH), 7.15-7.04 (3, 3H, Ph), 6.93 (dd, 2H, Ph, J = 1.46 Hz, J = 7.61 Hz), 6.17 (d, 1H, H-1′, J_1′,2′_ = 7.99 Hz), 4.82 (dd, 1H, H-2′, J_2′,3′_ = 2.49 Hz), 4.75, 4.72, 4.70, 4.67 (AB system, 2H, OCH_2_-3′, J = 12.02 Hz), 4.60, 4.57, 4.56, 4.53 (AB system, 2H, OCH_2_-4′, J = 11.91 Hz), 4.45, 4.42 (part A of AB system, 1H, OCH_2_-2′, J = 12.00 Hz), 4.37 (qd, 1H, H-5′), 4.26, 4.23 (part B of AB system, 1H, OCH_2_-2′, J = 12.00 Hz), 4.10 (t, 1H, H-3, J_3′,4′_ = 3.00 Hz), 3.64 (dd, 1H, H-4′, J_4′,5′_ = 5.11 Hz), 1.35 (d, 3H, J_6′,5′_ = 6.70 Hz). ^13^C NMR (100 MHz, CDCl_3_): δ 168.1 (CO), 153.4 (CqPh), 153.0 (C-2, C-4), 151.0 (C-6), 145.1 (C-8,Cq), 139.5, 139.4, 138.5 (CqPh), 138.5, 135.0, 133.9, 129.8, 129.5, 129.4, 129.3(5), 129.3, 129.2(5), 129.2, 129.1, 129.0, 128.9, 128.8(6), 128.7 (Ph), 125lç.1 (C-5), 81.3 (C-1′), 81.1 (C-4′), 76.0 (C-3′), 74.6 (C-2′), 74.1 (C-5′), 73.8 (OCH_2_-3′), 73.3 (OCH_2_-4′), 72.4 (OCH_2_-2′), 18.3 (C-6′). HRMS (CI^+^); calcd for C_39_H_37_N_5_NaO_5_ (M+Na)^+^: 678.2687, found: 678.2695.

Data for compound **16** using **14** as starting material: Syrup. Yield = 36%. Rf = 0.7 (DCM/EtOAc 1:1). [α]_D_^20^ = −29 (*c* 1, MeOH). ^1^H NMR (400 MHz, CD_3_OD): δ 8.55 (s, 1H, H-2), 8.32 (s, 1H, H-8), 8.10 (d, 2H, *ortho* PhCO, J = 6.91 Hz), 7.63 (t, 1H, *para* PhCO, J = 7.30 Hz), 7.54 (t, 2H, *meta* PhCO, J = 7.50 Hz), 7.41 (d, 2H, Ph, J = 7.49 Hz), 7.35-7.23 (m, 9H, Ph, NH), 7.01 (dd, 3H, Ph, J = 2.00 Hz, J = 5.11 Hz), 6.88-6.83 (m, 2H, Ph), 5.92 (d, 1H, H-1′, J_1′,2′_ =0.72 Hz), 4.94, 4.91 (part A of AB system, 1H, OCH_2_-4′, J = 11.05 Hz), 4.83, 4.80 (part A of AB system, 1H, OCH_2_-3′, J = 12.05 Hz), 4.76, 4.73 (part B of AB system, 1H, OCH_2_-3′, J = 12.05 Hz), 4.72, 4.69 (part B of AB system, 1H, OCH_2_-4′), 4.69, 4.66 (part A of AB system, 1H, OCH_2_-2′, J = 11.79 Hz), 4.29, 4.27 (part B of AB system, 1H, OCH_2_-2′, J = 11.79 Hz), 4.23 (br s, 1H, H-2′), 3.95 (dd, 1H, H-3′, J_3′,2′_= 2.47 Hz, J_3′,4′_ = 9.21 Hz), 3.77-3.59 (m, 2H, H-4′, H-5′), 1.36 (d, 3H, H-6′, J_6′,5′_ = 5.87 Hz). ^13^C NMR (100 MHz, CD_3_OD): δ 167.9 (CO), 153.8 (C-2), 151.7 (C-4), 150.8 (C-6), 144.0 (C-8), 139.6, 139.5, 138.1, 135.0 (PhCq), 129.8, 129.6, 129.5, 129.4, 129.3, 129.2, 129.1, 128.9, 128.8(7), 128.8(5), 128.8 (Ph), 124.2 (C-5), 84.0 (C-3′), 83.2 (C-1′), 80.6 (C-4′), 76.3 (C-5′), 76.2 (OCH_2_-4′), 75.5 (OCH_2_-2′), 74.5 (C-2′), 73.8 (OCH_2_-3′), 18.5 (C-6′). HRMS (CI^+^); calcd for C_39_H_37_N_5_NaO_5_ (M+Na)^+^: 678.2683, found: 678.2687.

### 4.2. Cholinesterase Inhibition Assay

The test compounds were assayed for their inhibitory activity toward *ee*AChE (electric eel AChE) and *eq*BChE (horse serum BChE), respectively, following Ellman’s method with modifications [21,22]. The BChE activity was determined in a reaction mixture containing 20 µL of *eq*BChE solution (0.9 U/mL in 0.1 M pH 8.0 phosphate buffer, PB), 20 µL of 5,5-dithio-bis-(2-nitrobenzoic) acid solution (DTNB 3.3 mM in 0.1 M pH 7.0 PB, containing 0.1 mM NaHCO_3_), 20 µL of a solution of test compound (five to seven concentrations, ranging 1 × 10^−5^ to 1 × 10^−9^ M (in 0.1 M pH 8.0 PB), and 120 µL of 0.1 M PB (pH 8.0). After 20 min of incubation at 25 °C, 20 µL of butyrylthiocholine iodide (BTC, 0.05 mM in 0.1 M pH 8.0 phosphate buffer, PB) was added as the substrate, and the hydrolysis rates of the substrate monitored at 412 nm for 5.0 min at 25 °C. For compounds **15** and **16**, this procedure was adapted to a final compound concentration of 5 μM. The concentration of compounds that produced 50% inhibition of BChE activity (IC_50_) was calculated by nonlinear regression of response/concentration (log) curve by using Prisma Graph Pad software (vers. 5.01). AChE inhibitory activity was determined similarly by using a solution of AChE (0.9 U/mL in 0.1 M pH 8.0 phosphate buffer, PB) and acetylthiocholine iodide (ATC, 0.05 mM in 0.1 M pH 8.0 phosphate buffer, PB) as substrate. The inhibition data are reported as means of IC_50_’s determined at least in three independent measurements.

### 4.3. UV-Visible Chelating Studies

UV-Visible spectroscopy offers the advantage of high sensitivity towards small changes that affect the electronic properties of ligand receptors [27], so absorption spectra of all reported compounds have been evaluated by recording UV-visible properties in subsequent conditions: 1.5 mL 1 × 10^−4^M of the probe are added to 1.5 mL 2 × 10^−5^M of each ion chloride salt’s solution (Copper Chloride (CuCl_2_), Zinc Chloride (ZnCl_2_), Iron (II) Chloride (FeCl_2_) and Aluminum Chloride (AlCl_3_)). A mixture of distilled water/DMSO 95:5 has been selected as a solvent for compounds **5**–**10** and **12**. For compounds **14**–**16**, the selected solvent was a mixture of distilled water/DMSO 90:10 due to poor water solubility. Furthermore, a titration assay of each compound has been carried out by adding increasing concentration of selected ion (final ion concentration = 1 µM, 2 µM, 3 µM, 5 µM, and 10 µM). 

### 4.4. Computational Studies

The initial geometries were optimized in Gaussian16 with CAM-B3LYP/6-31G(d) and integrated equation formalism polarized continuum model (IEFPCM) for water [28,29,30]. Final geometries were optimized with CAM-B3LYP/def2TZVP and IEFPCM for water. The vibrational frequencies were checked for true minima in all cases (no imaginary frequencies were found). To accelerate the search for the best configurations, the molecules were split into two parts: the sugar unit and the purine moiety. With respect to the purine moiety, the geometry optimization started from some initial purine geometries with small differences in the dihedral angle between the rings, and the optimized geometries always ended up in two conformers. The geometry optimization for the glycosyl groups followed the same procedure. The optimization was started from a series of different initial geometries for each sugar unit, and the most stable conformer was selected. For the final geometry optimizations, the best glycosyl conformer found was coupled with the two purine conformers, and a full optimization of the whole molecule was performed. 

NMR parameters were evaluated from single-point calculations with CAM-B3LYP/def2TZVP and IEFPCM for water. The computed NMR shielding tensors were converted to chemical shifts with empirical scaling factors determined from a test set of 80 molecules at the same level of theory [31].

## Data Availability

Data is contained within the article and Appendix A.

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
