# Peer review of "Exploring Mannosylpurines as Copper Chelators and Cholinesterase Inhibitors with Potential for Alzheimer’s Disease"

_pharmaceuticals, 2022, doi:10.3390/ph16010054_

Round 1

Reviewer 1 Report

This article is exploring mannosylpurines as copper chelators and choline esterase inhibitors as potential therapeutics against Alzheimer’s disease. Article is covering almost all data available up today on this class of inhibitors as potential drug candidates. 

The manuscript is decorated with 10 figures, 3 tables and 3 schemes describing synthetic pathways and conformations confirmed by NMR data and computational studies. 

The specific format of this very important article is exclusively designed and specifically classified by the selective copper chelating properties over zinc, aluminum and highlighting molecular conformations and the chelating molecular site. Additionally, is highlighting the role of sugars by tuning bioactivity and selectivity. This will constitute the important goals and novelty of this manuscript!

            The following suggested changes and recommendations should be introduced before the publication of the manuscript.

1.     Page 2, line 47, inserts “extremely” before “critical role”. 

2.     Page 3, figure 1 should be moved to page 2 at line 84. This will direct reader to important role of copper chelating drugs discussed in this paragraph. 

3.     Page 5, legend for scheme 1 should be inserted on page 4 directly under the scheme. 

4.     Page 7, line 238 the abbreviation hBChE should be consistently used in the text and as opposed with BChE, which is also human choline esterase. 

5.     Page 14, line 364, inserts “ (1C4 conformer)” after 16

6.     Page 18, line 587, the reference number [27] is in red color, should be corrected to black color. 

7.     Page 19, line 608, “couple” should be corrected to “coupled”.

 The manuscript is of good quality and importance and is well written and edited in order to meet the standard for the articles published in Pharmaceuticals. Thus, I certainly recommend it for publication after the correction of these suggested minor changes. 

Author Response

Dear Reviewer 

I attach the answer, all your comments have been followed and we thank you so much for your comments about our paper.

I take this opportunity to send you my season greetings!

With my best regards

Amelia Pilar Rauter

Reviewer 2 Report

Excellent manuscript from all possible perspectives: conception, development, methodologies, discussion, results and conclusion. The synthesis, NMR, computations, and bioactivity measurements are well carried out, with rigor. Excellent combination of methods for a focused scientific problem.

Author Response

Dear reviewer

We thank you so much for your comments and were very happy when we read them!

With my best regards

Amelia Pilar Rauter

Reviewer 3 Report

In the submitted manuscript, Schino et al. synthesized new series of compounds, mannosylpurines. They investigated their inhibitory activity toward two cholinergic enzymes, acetyl, and butyrylcholinesterase. Additionally, their chelating effects toward Cu2+, Fe2+, Zn2+, and Al3+ were spectrophotometrically investigated. I have to stress that the synthesis of compounds, as well as biological and chelating tests, were well done. However, in this form, this manuscript needs minor revision, and I have listed these issues and recommendations below:

1) All Figure captions have to be improved so that they include more details about experimental conditions.

2)   The resolution of Figures 3, 9, and 10  ( some of the Figures related to UV/Vis) has to be improved.

3) I wonder why the authors did not describe the assays linked with human cholinesterase enzymes in the Material and Methods section.

Finally, by performing the proposed corrections, the manuscript will satisfy the standards of this journal, and I suggest acceptance.

Author Response

Dear reviewer,

Thank you so much for your comments, whcih were fully followed by us.

With my best regards

Amelia Pilar Rauter
